# Conversion of DNA Sequences: From a Transposable Element to a Tandem Repeat or to a Gene

**DOI:** 10.3390/genes10121014

**Published:** 2019-12-05

**Authors:** Ana Paço, Renata Freitas, Ana Vieira-da-Silva

**Affiliations:** 1MED-Mediterranean Institute for Agriculture, Environment and Development, University of Évora, 7002–554 Évora, Portugal; ana.vieiradsilva@gmail.com; 2IBMC-Institute for Molecular and Cell Biology, University of Porto, R. Campo Alegre 823, 4150–180 Porto, Portugal; Renata.Freitas@ibmc.up.pt; 3I3S-Institute for Innovation and Health Research, University of Porto, Rua Alfredo Allen, 208, 4200–135 Porto, Portugal; 4ICBAS-Institute of Biomedical Sciences Abel Salazar, University of Porto, 4050-313 Porto, Portugal

**Keywords:** tandem repeats, dispersed sequences, origin of tandem repeats, mobilization of tandem repeats, DNA remodeling mechanism

## Abstract

Eukaryotic genomes are rich in repetitive DNA sequences grouped in two classes regarding their genomic organization: tandem repeats and dispersed repeats. In tandem repeats, copies of a short DNA sequence are positioned one after another within the genome, while in dispersed repeats, these copies are randomly distributed. In this review we provide evidence that both tandem and dispersed repeats can have a similar organization, which leads us to suggest an update to their classification based on the sequence features, concretely regarding the presence or absence of retrotransposons/transposon specific domains. In addition, we analyze several studies that show that a repetitive element can be remodeled into repetitive non-coding or coding sequences, suggesting (1) an evolutionary relationship among DNA sequences, and (2) that the evolution of the genomes involved frequent repetitive sequence reshuffling, a process that we have designated as a “DNA remodeling mechanism”. The alternative classification of the repetitive DNA sequences here proposed will provide a novel theoretical framework that recognizes the importance of DNA remodeling for the evolution and plasticity of eukaryotic genomes.

## 1. Introduction

Eukaryotic genomes contain a high diversity of repetitive DNA sequences [1,2]. The amplification/deletion of these sequences contributed significantly to the extraordinary variation in genome size found between taxa [3,4,5,6,7]). In the animal kingdom, the genome size could vary from 20 Mb to 130 Gb, which is mainly due to differences in the content of repetitive sequences [8]. The same large variation was identified in plants. For instance, the fraction of repetitive genomic DNA is 13–14% (125 Mb–157 Mb) in *Arabidopsis thaliana* but, contrastingly, is 77% (2.5 GB) in *Zea mays* [9].

The biological role of this repetitive DNA fraction has been a topic of great interest namely to understand evolution and disease [10,11,12,13]. In summary, this fraction seems to be involved in DNA packaging, the evolutionary events of the genome (through promoting DNA instability), gene expression, and epigenetic mechanisms [14,15,16,17,18,19,20,21,22,23,24,25]. The analysis of the DNA sequences located in chromosomal breakpoint regions strongly suggests that repetitive sequences work as a driving force in the occurrence of chromosomal rearrangements, since these regions are extremely rich in these sequences [26,27]. The repetitive nature, per se, of the different classes of repeats located in these breakpoint regions favours recombinational events between homologous sequences in non-homologous regions, which may culminate in chromosomal restructurings [21]. Besides, an analysis devoted to the transcriptional activity of some repetitive sequences input a role of these sequences in control of gene expression, cellular response to stress and centromeric function, specifically by RNA interference mechanisms [28].

Despite the increasing evidence pointing to a functional significance of the repetitive DNA fraction, there are still limitations to characterizing their role in different biological processes due to their high diversity, genomic abundance, complex evolution mode, and difficulty in isolation and sequencing [29,30,31]. Thus, the real biological relevance of this fraction of eukaryotic genomes is yet to be revealed.

This review presents a compilation of evidence on the organization and evolutionary relationship between different classes of repetitive sequences, and between repetitive sequences and genes. This evidence shows that the classification of repetitive sequences based on its genomic organization as “in tandem” or “dispersed repeats” is not “as black and white”, which reinforces the need for an updated classification. Further, this evidence also shows a high remodulation of the repetitive sequences in the eukaryotic genomes, which could culminate in the origin of a new sequence type, namely genes. This new way to face the classification and study of the repetitive sequences will contribute to a better understanding of genomic plasticity and its contribution to eukaryotic species evolution and adaptation to environment.

## 2. Tandem Repeats and Dispersed Repeats: Is Their Organization So Different?

Repetitive sequences are classified in two classes, tandem and dispersed repeats, according to the genomic organization of their copies (Figure 1) [32,33]. In turn, each class is divided into several subclasses or families [3,4,5,6,34,35,36,37]. With the goal to facilitate the annotation and classification of the repetitive elements in eukaryotic genomes, an open access database for repetitive sequence families (Dfam) was built [37,38].

### 2.1. Tandem Repeats Organization

Traditionally, tandem repeats have been structurally characterized by a sequential arrangement of repeat units, positioned one after the other in two possible repeat orientations, head-to-tail repeats (direct repeats) or head-to-head repeats (inverted repeats) [33]. Excluding the genic repetitive DNA sequences (e.g., ribosomal genes), three distinct subclasses are mainly recognized, namely microsatellites, minisatellites, and satellites (satellite DNAs—satDNAs; nongenic tandem repeats) (Figure 1).

It is generally considered that the main difference between micro, mini, and satellites relates to the length of their array of repeat units in a chromosomal location [33,39,43]. Micro and minisatellites are classified as short tandem repeats, composed by shorter arrays of copies: ranging from 10 to 100 repeat units for microsatellites, and up to 100 repeat units for minisatellites [39,44]. However, it is important to mention that there is no consensus on the definition of microsatellites and minisatellites, since there is also no consensus in the repeat unit size, or in the minimal number of repeat units in an array of copies differentiating micro and minisatellites. The threshold for repeat unit size varies between six and 10 nucleotides [33,39,45].

The SatDNAs have traditionally been considered as organized into long arrays, with millions of copies and, thus, have been named as long tandem repeats [39,46]. However, satDNAs with a different organization have already been reported. Louzada and colleagues (2015) [47], identify a satDNA (PMsat) with high sequence conservation in the genomes of five rodents. However, the PMSat is not always organized into long array repeat units, even in species where it is highly abundant, as is the case of *Peromyscus maniculatus bairdii*. In this species the authors found short PMSsat arrays and even dispersed isolated monomers, which reinforces that the limitation of techniques to isolate and analyse the repetitive fractions of genomes (as sequencing, assembly and mapping technologies) is a determining factor for the study of repetitive sequences, as well as for its accurate classification. Other reports have also come to prove the same, showing a dispersed genomic organization of short arrays of satDNA repeat units [44,48].

Besides the length of arrays, the repeat units of satDNA (monomers) can also show a great variation in size, ranking from five nucleotides, as in human satellite III [49], which is similar to a micro or a minisatellite, but with longer arrays of repeat units, up to several hundred base pairs as in the *Microtus* MSAT-2570 [50]. However, for plants and animals, the most common length is 150–180 bp and 300–360 bp, respectively, which is believed to be associated with the requirements of DNA length wrapped around one or two nucleosomes [11,51,52].

The genomic distribution presented by micro, mini, and satDNAs is also traditionally considered distinct in the literature. Generally, both microsatellites and minisatellites are distributed throughout the genome (dispersed), in both euchromatic and heterochromatic regions [33,45]. Nevertheless, minisatellites are also characterized by their accumulations in (sub)telomeric regions [43,53]. The satDNAs are mainly located in heterochromatic regions of the chromosomes, thus satDNAs are preferentially found in and around centromeres [54,55,56,57]. However, once again, exceptions have been reported as to what is generally assumed. Indeed, satDNA can also be located at interstitial and terminal positions of chromosomes [26,47,48].

### 2.2. Dispersed Repeats Organization

Dispersed repeats are mainly represented by transposable elements (TEs). With the recent proliferation of genomic sequencing studies, TEs have emerged as highly diverse, ubiquitous and abundant genomic elements, constituting approximately half of the human genome and up to 95% of DNA in plants [58,59,60,61]. The diversity of TEs reflects their evolutionary mode [62]. By the accumulation of mutations, TEs generate new families and subfamilies, “escaping” to selection [63]. Despite this diversity, it was possible to group them into two major classes according to their modes of transposition (mobilization in genomes): retrotransposons (RE, class I elements) and DNA transposons (class II elements) (Figure 1) [60,61].

Traditionally, the dispersed repeats consist of sequences represented several times in the genome, whose copies are not clustered or are organized in short clusters, presenting a wide distribution throughout the genome [39,43,64]. However, some works show dispersed repeats with an accumulated genomic organization [65,66,67,68,69,70], which can lead to the supposition of some tandem organization for these sequences. As referred to previously, the genomic accumulation of tandem repeats is common in certain genomic locations, such as the telomeric sequences (microsatellites of (TTAGGG/CCCTAA)_n_) in the genomes of all mammals at telomeric and interstitial positions [71,72,73], and satDNAs at centromeric regions [57]. In fact, different works have specifically explored the organization of tandem repeats at telomeric and centromeric regions [26,66]. Nevertheless, how TE blocks accumulated in certain regions of the genome are in fact organized is not reported in many studies. Only a few works have revealed that TE clusters could be interrupted by other TEs or also by genes, where there might also be a short tandem arrangement of some TEs [67,69].

### 2.3. Repetitive Sequences: A New Classification Based on the Presence or Absence of Retrotransposons/Transposon Specific Domains

In this review, we suggest an alternative to the traditional classification of DNA repetitive sequences mainly based on the genomic organization of its copies. We suggest that the repetitive sequences should not be divided as tandem repeats and dispersed repeats, once repeats organized in tandem could also present a dispersed organization, and vice-versa. As referred to previously, several tandem repeats have a dispersed organization in distinct genomes. In contraposition, some TE copies may be positioned one after the other in tandem organization, presenting some chromosomal regions several complete or incomplete copies of a specific TE. As such, these sequences should be classified regarding other characteristics, namely the presence or absence of retrotransposons/transposon specific domains in the sequence of its copies. Therefore, the repetitive sequences could perhaps be classified as repeats presenting retrotransposons/transposon specific domains or repeats not presenting retrotransposons/transposon specific domains.

## 3. Repetitive DNA Remodelling

Different studies show that a repetitive element can be remodelled into a different sequence, repetitive or not, which proves an evolutionary relationship among DNA sequences, and suggests that the evolution of the genome is characterized by a frequent repetitive sequence reshuffling, a process that we have called a “DNA remodelling mechanism”. In fact, some authors show that the tandem repeats could have their origin in TEs [74], and that satDNAs could also evolve to coding sequences [75].

### 3.1. Transposable Elements in Origin and Genomic Distribution of Micro, Mini and Satellite DNAs

Sequence similarities between tandem repeats and TEs [76,77,78] indicate a strong evolutionary relationship between these repetitive sequences. In addition, it is also believed that TEs are involved in the origin of some sequence motifs that characterized some satDNAs, as the CENP-B box, presenting this sequence motif strong similarity with the terminal inverted repeats of pogo transposons [79]. In fact, computer simulations have suggested that satDNA monomers could be generated from a wide variety of non-satellite sequences and propagated into an array by unequal crossing-over [80]. These non-satellite sequences are often TEs [26,31,76,81,82,83,84,85,86,87,88]. In Table 1, different examples known in the literature are listed where TEs or parts of TEs were converted into other repetitive sequence or altered genes.

The evidence for TE conversion into new non-coding repetitive sequences or genes are reinforced by the fact that, in humans and mice—the first fully-sequenced genomes—it was estimated that the repetitive DNA derived from TEs comprises from 40% to almost half of these genomes [110,111]. These values could be quite underestimated, with substantial amounts of older sequences not being detected due to their already highest divergence compared to the consensus sequences used for their detection. Ahmed and Liang (2012) [107], for example, considered that the ability of TEs to contribute to genome expansion is due, not only to retrotransposition (increasing its copy number in the genomes), but also by generating tandem repeats.

The exact mechanisms underlying the origin/expansion of tandem repeats from TEs are not yet completely known, but probably involved more than one mechanism. A tandem repeat sequence arises after amplification events followed by subsequent molecular mechanisms. It is widely accepted that the first repetitions of microsatellites may have originated by chance, and then expanded by slipped-strand mispairing, as proposed by Levinson and Gutman (1987) [112]. However, some studies suggested that TEs contain one or more sites predisposed to the formation of microsatellites. The Poly(A) tract at the 3’ end of mammalian non-LTR retrotransposons (autonomous LINEs/nonautonomous SINEs) provides a susceptible site to reverse transcription errors, which could lead to the genesis of A-rich microsatellites [113]. The description of microsatellites located at the 5’ end and internal regions of retroelements is also available in the literature [100].

Regarding minisatellites, several reports point to their origin from a variety of TE families and subfamilies [103,104,105,106,108], namely from nonautonomous non-LTR retrotransposons as Alu and B1 SINE elements [105,106] or from LTR retrotransposons [103,104,108]. According to Haber and Louis (1998) [114], the origin (initial event) of the first repetitions of some minisatellites appears to have been mediated by replication slippage or unequal crossing-over, involving very short repeats (5–10 bp) that flank a motif which will be amplified as the repetition unit of these minisatellites (Figure 2). The evidence that repetitive elements, such as Alu elements, commonly have short direct repeats in their sequences, makes them very prone to the origin of minisatellites by this mechanism [106]. Subsequently, the amplification of the duplicated motifs into a minisatellite array could then occur by additional replication slippage events, gene conversion, or by unequal crossing-over between the longer homologous regions [114]. This mechanism seems plausible to explain the origin of minisatellites; however, it cannot explain the origin of the satDNA with larger repeat units, due to the distance over which the initial event must have occurred (replication slippage or unequal crossing-over involving flanking short repeats). Nevertheless, a very similar mechanism was proposed to explain the origin of satDNAs, as for maize centromeric repeats, with most of their monomers presenting more than 700 bp [87].

Other interesting theories on the origin of satDNAs from TEs have been suggested. Wong and Choo (2004) [74] proposed the “first steps” hypothesis for the origin of satDNA repetitions, based on the duplication of part of a TE sequence by unequal crossing-over between homologous TE elements, which could be in the same or in different chromosomes (Figure 3). Once a tandem repetition of full or partial TEs is generated in a genome, the expansion of these novel repeat units can slowly occur over time. Mutational changes, followed by successive rounds of crossing-over homogenization (concerted evolution of tandem repeats), can justify the divergence observed between the emergent satDNA and the original TE, presenting only conserved parts of their sequences [74]. This mechanism is recurrently used to explain the origin of satDNAs. An example is the work of Dias et al. (2015) [88], suggesting the emergence of a satDNA from central tandem repeats of a helitron (DINE-TR1) in *Drosophila* species.

The DNA transposons, or specifically their transposase activity, have been also referred to in the birth of sequence duplications. Kapitonov and Jurka (1999) [84] propose that the breaks induced by transposases during transposition (endonucleolytic tranposase activity) could favour recombination processes in order to repair the double strand breaks. This event may originate the first repetitions of a tandem repeat, which could afterwards be amplified in a large array of copies.

Beyond the role suggested in the origin of the tandem repeat, the TEs were also implicated in its relocation/distribution throughout the genome [31,54]. It is logical to believe that when tandem repeats are included within the mobile element sequence (for example, when the tandem repeats have its origin by duplication of part of TE sequence), maintaining the competence for mobilization for these TEs. The transposition mechanism can easily disperse tandem repeats throughout the genome (Figure 4A). This hypothesis is commonly accepted for short tandem repeats as micro and [115,116,117], which present short arrays of copies compared to satDNAs [39]. In fact, a considerable part of micro and minisatellites in eukaryotic genomes are embedded within mobile elements [115,117], which points to an important role of TEs in its genomic distribution, explaining its common dispersed chromosomal location. Moreover, TEs are present in pericentromeric regions of a wide range of species [27,70,74], being these regions also mainly built by satDNAs, which certainly facilitate the dispersion of these highly tandem repeats by retrotransposition. 

Regarding LINE-1 retrotransposons, several reports suggest a location of these elements in pericentromeric regions of different mammalian species chromosomes [27,66,70,118,119,120], pointing to an intermingling of these retrotransposons with satDNAs [66,118]. This complex organization pattern of repetitive sequences in the pericentromeric regions eventually favours the dispersion of satDNAs to other genomic locations by LINE-1 retrotransposition, since these elements frequently allow for the transduction of flanking non-LINE-1 DNA to new genomic locations (Figure 4B). This transduction is a consequence of a LINE-1 incorrect retrotransposition process [121,122,123]. Sometimes by retrotransposition, the TE sequences, along with its adjacent DNA, are copied and subsequently integrated into another genomic locations. This results in the duplication and genomic dispersion of the TE flanking sequences [124], as in, for example, satDNA monomers.

Furthermore, we can further speculate that DNA transposons can also allow the transduction of tandem repeats sequences [61], in a process similar to the one recognized in bacteria, for the transference of genes (e.g., antibiotic resistance genes) within and between bacterial genomes [125,126]. Sequences flanked by two “cut-and-paste” transposons can probably be mobilized in a genome, when its transposases use Terminal Inverted Repeats (TIRs) of the two different transposons to induce breaks for the mobilization (Figure 5A). If the TIRs of each DNA transposon are exclusively used, only these elements will be mobilized. Interestingly, as referred to previously, some similarity exists between the CENP-B box motifs and transposase recognition sites of DNA transposons [79], which may also lead to the identification of the CENP-B boxes as a break site for transposition [127]. Therefore, the common presence of CENP-B box motif in different satDNA families [127,128,129,130] can be involved in the mobilization of satDNAs copies during transposition [127]. No copies (monomers) of a satDNA described as presenting CENP-B boxes have these sequence motifs [131]. Thus, several monomers flanked by a DNA transposon and a CENP-box could be mobilized at the same time to another location, and subsequently amplified by different recombinational mechanisms (Figure 5B). This capacity of DNA transposons for the relocation of sequences flanked by them, or specifically by their TIRs, is indeed already used in medicine for gene therapy [132].

### 3.2. Repetitive Sequences in the Origin of Coding Sequences

Transposable elements and satDNAs could also be involved in the evolution of genes, but most interestingly in the origin of new genes or gene variants. The noticeable ability of the TEs to produce genetic mutations when integrating at new genomic sites was recognized more than 50 years ago [20,133]. Nevertheless, despite most of these insertions being either neutral or deleterious to their host, its inclusions into new locations may also be advantageous, promoting gene evolution and the codification of more efficient protein variants. One of the most publicized discoveries about this subject is the resistance to HIV-1 (Human Immunodeficiency Virus) infection in owl monkeys, which presents an altered *TRIM5* gene with a cyclophilin A domain acquired by LINE-1 retrotransposition [109]. The binding of this cyclophilin domain to the HIV-1 viral capsid leads to a disruption of the infection process [134]. However, these primates are permissive to other immunodeficiency virus, such as the simian immunodeficiency virus (SIV) [109].

Recently, some works have shed light on questions about the de novo origin of protein-coding genes (or variants) from non-coding DNA [75,135,136,137], such as satDNAs. It is believed that the origin of completely novel genes from non-coding DNA is an evolutionary process comprising two big steps. In the first step, the non-coding DNA sequences are transcribed and then acquire translatable open reading frames [135] (Figure 6). Some works have already reported open reading frames in the monomers of satDNAs [138,139], an important step that might have allowed these sequences to evolve into coding sequences.

## 4. Concluding Remarks

Pioneer studies on the eukaryotic genomic repetitive fraction has led to the classification of the repetitive sequences into two major groups according to the organization of copies within the genome: tandem repeats (as satDNAs) and dispersed repeats (TEs). Because of that, these sequences have been mostly investigated separately, with the important evolutionary relationships that exist between them not being considered. However, more precise genomic and bioinformatic analyses have now shown that these sequences do not have such a tight genomic organization. Some satDNAs may present dispersed isolated monomers in a genome [47] while TEs may have a kind of tandem organization of their copies [69]. Moreover, it became evident that a repetitive element can often be remodelled into a different sequence, a repetitive non-coding sequence or even a coding sequence. This suggests that the repetitive DNA elements in eukaryotic genomes seem to be in frequent remodulation, changing its organization and function. Therefore, an update in repetitive sequence classification is now mandatory. Here, we have proposed a new classification for these sequences, not based on the genomic organization of their copies, but on other sequence features, namely the presence or absence of retrotransposon/transposon specific domains in their copies. Thus, we propose two new groups for the classification of the repetitive sequence: repeats presenting retrotransposons/transposons specific domains and repeats not presenting retrotransposons/transposons specific domains. This new classification demonstrates more clearly the evolutionary relationship between these sequences, promoting also more works to study together sequences that were previously considered very distinct. Their joint study is indeed very important for a better understanding of their function in genomes, showing that the evolutionary relationship between these sequences and the way that they can convert to each other is highly associated to the evolution of the genomes themselves.

Accordingly, we believe that future combined studies regarding TEs and tandem repeats, namely concerning chromosomal location and molecular similarity, will increase our knowledge about the evolution of eukaryotic genomes. The combined studies of related repetitive sequences can help us to understand the reason for some evolutionary tracks of sequences, and to understand these tracks in such a way that this genome plasticity makes the eukaryotic species better adapted to environmental conditions.

## Figures and Tables

**Figure 1 genes-10-01014-f001:**
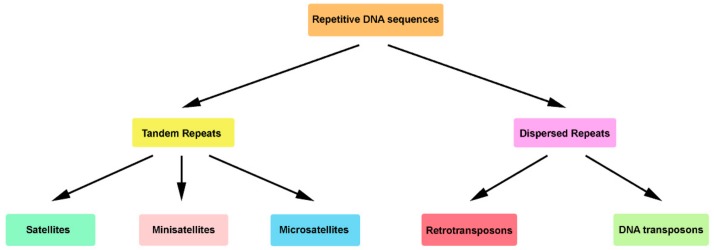
Repetitive DNA sequences in eukaryotic genomes. This schematization collects the information of several works [16,20,33,39,40,41,42]. Here, only the largest subclasses of tandem and dispersed repeats are represented, not including the genic repetitive DNA sequences families, as tandem paralogues genes, ribosomal genes (tandem organization), retropseudogenes, transfer RNA genes, and dispersed paralogues genes (dispersed organization).

**Figure 2 genes-10-01014-f002:**
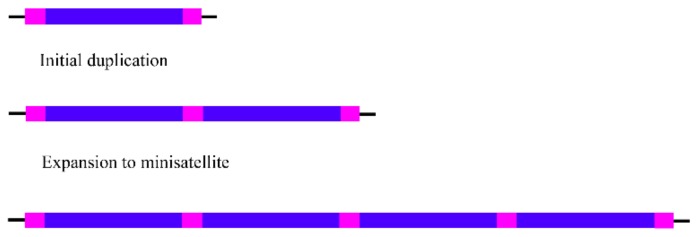
Initial event in the origin of the first minisatellites repetitions. Origin of a duplication by replication slippage or unequal crossing-over between short flanking repeats, followed by a subsequent expansion into a minisatellite.

**Figure 3 genes-10-01014-f003:**
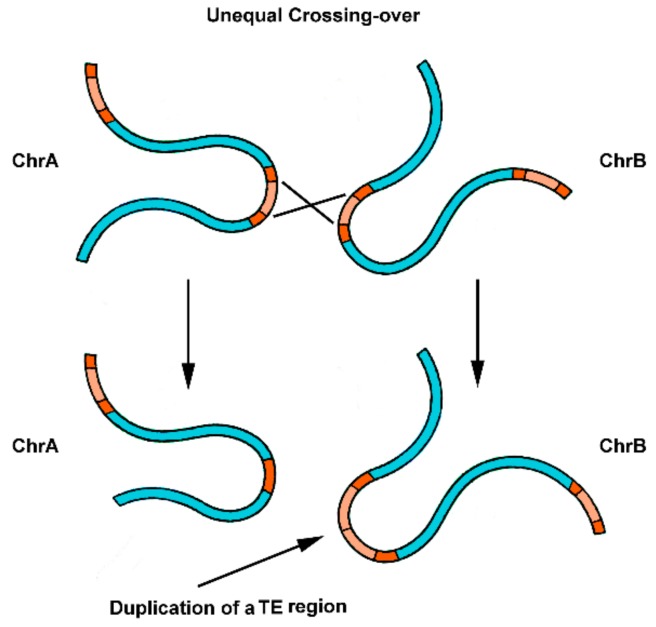
Initial steps for the origin of satDNA repeats from parts of a TE. The duplication of part of a TE sequence occurs by unequal crossing-over between homologous dispersed repeats present in chromosomes A and B (chrA and ChrB). The expansion of these novel repeat units can occur through time and result in a satDNA array of copies.

**Figure 4 genes-10-01014-f004:**
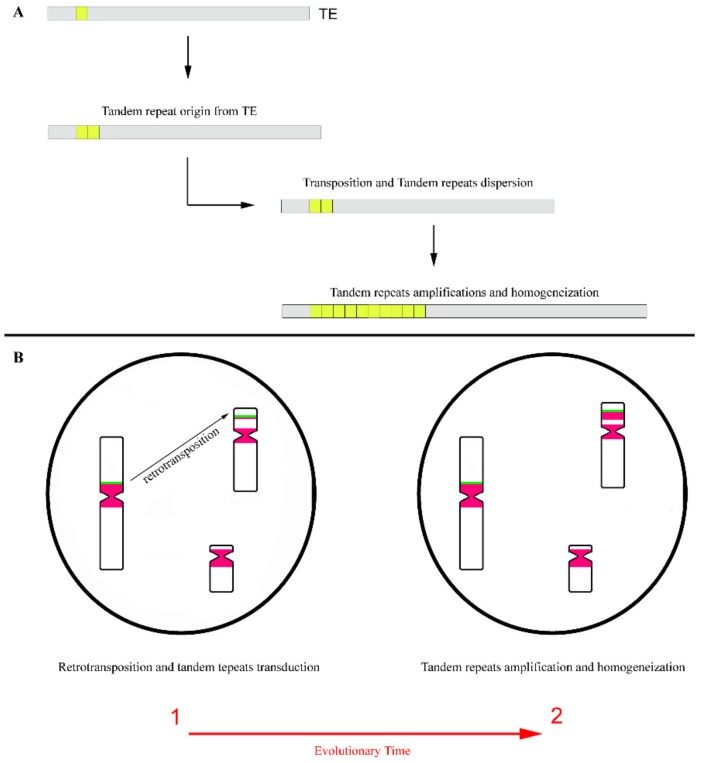
Dispersion of tandem repeats by transposition. (**A**) Origin of tandem repeats from a part of a transposable element (TE) and its dispersion by transposition. The first duplications of a tandem repeat were originated from a part of a TE. As these repeats are included in the TE sequence, could then be dispersed by transposition. After, these repetitions may be amplified and homogenized in an array of copies. (**B**) Transduction of tandem repeats flanking a retrotransposon and its consequent dispersion throughout the genome by retrotransposition. Retrotransposon evidenced by a green block and tandem repeats evidenced by pink blocks. During the evolutionary time, the tandem repeats that were moved to new chromosomal locations could be amplified and homogenized, originating arrays of copies in these locations.

**Figure 5 genes-10-01014-f005:**
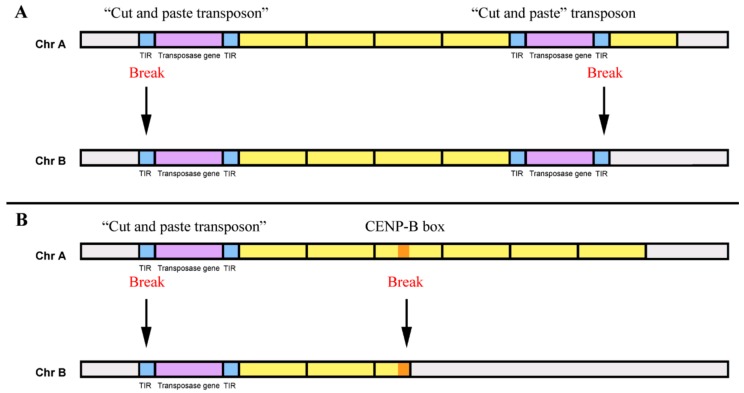
Dispersion of tandem repeats by “cut-and-paste” transposons. (**A**) Mobilization of sequences flanked by two “cut-and-paste” transposons. The breaks for the mobilization induced by transposases occur at the terminal inverted repeats (TIRs) of the two transposons. Yellow boxes: tandem repeats monomers, blue boxes: TIRs, violet boxes: transposase genes, grey boxes: remaining sequences of the chromosomes A and B. Chr: chromosome. (**B**) Mobilization of sequences flanked by a “cut-and-paste” transposon and a CENP-B box. The breaks for the mobilization occur at the TIRs of a transposon and a CENP-B box. Yellow boxes: tandem repeats monomers, blue boxes: TIRs, violet boxes: transposase genes, Orange box: CENP-B box, grey boxes: remaining sequences of the chromosomes.

**Figure 6 genes-10-01014-f006:**
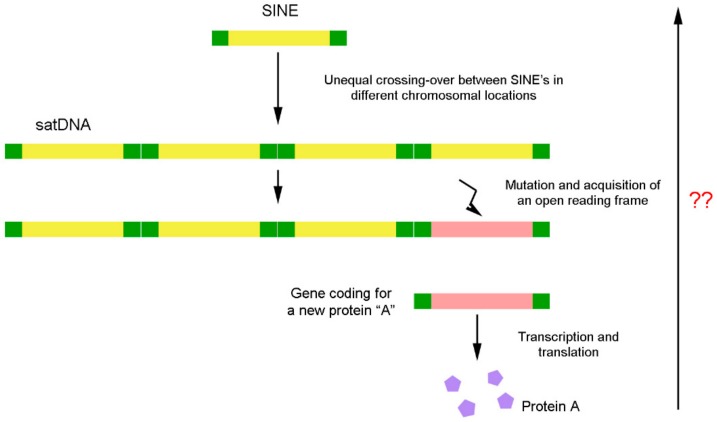
DNA remodelling process. Evolution of a satellite DNA sequence from a transposable element and its subsequent conversion in a coding sequence. The reverse sense of the process was not proved yet. ??- Is up to now unknown if occurs the opposite sense of this process for DNA sequences evolution.

**Table 1 genes-10-01014-t001:** Transposable elements in the origin of other repetitive sequence or altered genes.

Transposable Element	New Sequence or New Sequence Variable	Reference
SINE-like elements	Satellite 1 of *Xenopus leavis*	[89]
LTR retrotransposons	RPCS satDNA of *Ctenomys* rodents	[81]
SINE-like elements	Hy/Pol III satDNA european salamander	[82]
pDv mobile element	pvB370 satDNA of *Drosophyla virilis*	[83]
LINE-1 elements	Common cetacean satDNA	[76]
TART and HeT-A retrotransposons	18HT satDNA of *Drosophila melanogaster*	[90]
Atenspm2 transposons	Ensat1 of *Arabidopsis thaliana*	[84]
Crwydryn retrotransposon	E3900 satDNA of rye	[91]
MITE elements	D1100 satDNA of rye	[91]
SGM-IS transposons	SGM satDNA *Drosophila guanche*	[92]
Ty3/gypsy-retroelement	250 elements of satDNA of wheat	[93]
MITE-like elements	Xstir satDNA of *Xenopus leavis*	[94]
MITE elements	*Hin**dIII* satDNA of oysters	[95]
Sore1 retrotransposon	Sobo satDNA of potatoes	[96]
CR1 retrotransposons	*Hin**fI* satDNA of chicken	[97]
Ty3/gypsy-like ogre elements	PisTR-A satDNa of pea	[31]
MITE-like elements	BIV160 satDNA of bivalves	[77]
CR1-C retrotransposons	Cen2, 3, 4, 7 and Cen11 satDNAs of chicken	[98]
CRM1 and CRM4 retrotransposons	CRM1TR satDNA of maize	[87]
Helytrons elements	CTRs satDNA of *Drosophyla*	[88]
LINE-1 elements	PROsat of *Phodopus roborovskii*	[26]
*Alu* elements	A-rich primates’ microsatellites	[99]
SINE elements	BARE-1, WIS2-1A and PREM1 microsatellites of barley	[100]
LINE-1 elements	A-rich mammalian microsatellites	[101]
*Alu* elements	(GAA)n human microsatellite	[102]
MITE elements	GTCY(n) microsatellites of insects	[86]
*Alu* elements	pλg3 human minisatellite	[103]
MaLR retrotyransposon	Ms6-hm mouse minisatellites	[104]
SINE B1 elements	(GGCAGA)n mouse minisatellite	[105]
*Alu* elements	(CGGGAGGC)n human minisatellite	[106]
*Alu* elements	Minisatellites of human	[107]
Gmr9/Gm ogre retrotransposons	Gmr9-associated minisatellites of soybean	[108]
LINE-1 elements	TRIM5 gene with a cyclophilin A domain	[109]

SINE: Short interspersed nuclear element, LTR: Long terminal repeats retrotransposons, LINE: Long interspersed nuclear element, MITE: Miniature inverted repeat transposable elements, satDNA: Satellite DNA.

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
