# Peer review of "Conversion of DNA Sequences: From a Transposable Element to a Tandem Repeat or to a Gene"

_genes, 2019, doi:10.3390/genes10121014_

Round 1
Reviewer 1 Report
The genomes of eukaryotic organisms contain a large number of repetitive sequences, which are mainly composed of tandem and interspersed repeats. These repetitive sequences play an important role in genome evolution. However, due to the diversity of their structure, any classification method is difficult to include all the real situations. Based on the analysis of the existing research results, the authors put forward that there are similarities between the tandem repeats and the interspersed repeats with respect to the genome structure, and suggest that the tandem repeat and the dispersed repeat should be combined into one group. Although I don't fully agree with this classification method, this suggestion provides an interesting view to optimize the repetitive sequence classification and analysis. In my opinion, the authors need to further analyze and discuss the following aspects.
The combination of tandem and interspersed repetitive sequences into variable copy sequences is too simple and not specific. Although there is evidence to support the scattered distribution of satellite DNA and the tandem distribution of transposons, such classification only considers its distribution patterns in the genome, and does not consider the structural characteristics of the sequence itself. For example, transposons contain the whole or part of the conservative domains specific to transposons. Moreover, it is difficult to assemble repeat sequences perfectly based on the current sequencing technology. The reliability of copy number analysis only based on genome information needs further experimental verification. The similarity between transposons and tandem repeats indicates that they can convert mutually. Some satellite DNA originated from transposons. Different types of repeats and the transformation from repeats to genes can occur, but whether all of them are bidirectional or cyclic is questionable. Thus, I think it is more proper to use “conversion” than “recycling”. Lines 575-579, the two references are the same.
Author Response
First, we would like to thank the reviewer for the detailed review of this article, and all the attention that you devote to it. His/her comments greatly contributed to the improvement of this paper. In fact, when we read your suggestions, we agree with all of them:
We agree that the classification of tandem and interspersed repetitive sequences into a single group is not a progress on its classification. Despite the evidences supporting the scattered distribution of the micro, mini and satellite DNAs and the tandem distribution of retrotransposons or transposons, the classification of repetitive sequences in a single group, based only on its organization, is not correct. We agree completely with a classification based on other sequence features, considering the presence or absence of retrotransposons/transposons specific domains, as you suggest. We agree that the term "conversion" is more correct than "recycling". In fact, we have no evidence of a bidirectional or cyclic process, although repetitive sequences can be converted into other sequence types, such as genes. Therefore, we changed the term "recycling" to "conversion" or "remodelling" throughout the text. We deleted the duplicated reference.
Reviewer 2 Report
In this review, the authors propose to update the classification of repeated genomic DNA into categories of tandem repeats versus dispersed repeats into a general class of "variable copy sequences". They propose this update on the basis of better knowledge of genome structure stemming from increase output of sequencing projects, which show a more nuanced picture of sequence origin and dynamics than the two categories (repeated or dispersed) woudl let think. The point of the authors is that classification into 2 categories has led researchers to study these sequences separately, which hinders ou comprehesion of their origin, dynamics and role in genome evolution.
Although I find the argument interesting, the manuscript suffers from several weaknesses that should be adressed.
The main text is composed of 2 parts. The first part aims at showing, through a review of current litterature, that there are some repeated sequences (satDNA in particular) that can exist both at the tandem repeat and the dispersed state. Reciprocally, they review some evidences that some dispersed sequences (of transposon origin) have evolved into tandem repeats.
To me, a few counter examples escaping the current classification does not necessarily justify to overhaul the current classification entirely. Given the evolutionary dynamics of these sequences, it is expected that multiple mechanisms may lead to their use in term of genome evolution, whether it is by change in copy number or genome localization. But in any case, I think the authors should make a better effort to illustrate their argument more clearly, with the help of clear figures, recounting some specific examples that they cite. In any case, since the repeat size is often a key into the mechanism that can lead to its amplification (transposition, recombination or replication defect), the main interest of the current classification is to relate these different mechanisms. The mechanistic aspects behind this classifiaction should be discussed more thoroughly. Reclassifying all repeats into one category independently of their mechanism of evolution but according to their purported common origin is not obviously a progress. The authors shoudl spend more time explaining why this woudl be important for future reserach aiming at understanding the functional role of these sequences.
The second part of the manuscript concerns the link between transposition and repeated DNA sequence, either by spreading of repeated DNA sequences by transposition or through repeated DNA sequences originating from transposons. The authors review a few evidences showing how transposons can drive repeated DNA sequence reshuffling into repeats, or even into coding genes. This is thought as an argument against the idea that repeated DNA is "junk DNA. Again, it seems that the community working on repeated DNA has already moved largely past this concept of junk DNA. Again, the review would gain in interest if the authors illustrated better their argument about sequence reshuffling in several detailed examples, and possibly with the help of a table listing the different examples known in the litterature where sequence reshuffling has been proposed, with solid evidence.
Finally, the english of the manuscript should be revised. Many sentences are gramatically incorrect. It would improve the reading of the manuscript if it was proofread by a native english speaker.
Author Response
First, we would like to thank the reviewer for the detailed review of this article, and all the attention that you devote to it. His/her comments made us improve our arguments and we believe that the ideas and concepts within the manuscript are now presented in a much more solid way.
In our opinion, the variety of studies showing a not so strict organization of repetitive sequences, as tandem or dispersed repeats, highlight the need for a better classification, able to better illustrate the evolutionary relationship between them and allowing their study as a unit. However, we also agree that this single-group classification is not a progress in the way of classifying repetitive sequences. Therefore, in this revised version of the manuscript, we propose a classification not based on the genomic organization of these sequences, but on motifs conserved within them, namely the presence or absence of specific retrotransposon/transposon domains. Thus, we propose two new groups for the repetitive sequence’s classification: repeats with retrotransposons/transposons specific domains (RTSD) or repeats with no retrotransposons/transposons specific domains (RTNSD).
For us, this new classification highlights more clearly the evolutionary relationship between these sequences, potentially inspiring more work addressing together sequences previously considered very distinct and unrelated. Their integrated study is indeed very important for a better understanding of their function in distinct genomes, showing that the evolutionary relationship between them and the way they can convert into each other is highly associated with the evolution of the genomes themselves. According to that, we believed that future combined studies, regarding TEs and tandem repeats, namely concerning its chromosomal location and molecular similarity, would increase our knowledge on the evolution of eukaryotic genomes. In addition, the combined studies of related repetitive sequences can help us to understand the reason for some evolutionary tracks and understand these tracks, in a way that this genomic plasticity makes the eukaryotic species more adaptable to environmental conditions.
Finally, we introduce in the revised version of the manuscript a table listing the different examples known in the literature, in which repetitive sequences were converted into other non-coding repetitive sequence or gene. Moreover, a new figure (Fig. 6) schematizing the “DNA remodelling mechanism” was also included in the revised version of the manuscript and text was proofread by a native English speaker.
Reviewer 3 Report
This paper is a complete, interesting and a very extens review about DNA satellites and transposable elements. The authors also propose a new classification of repetitive elements and suggest new mechanisms "new recycling mechanism" to explain the evolution of genoma based on repetitive sequences.
I recommend to define more accurately the new classification of repetitive elements (RE) and to compare it with the previus ones. In which sense it means a significant improvement?. Nowadays it is accepted, but unknown in many cases, the close relation between transposable elements(TE) and tandem repeats: sets of TEs that are located consecutively like tandem repeats (TR), and different TD distributed as TE.
The second proposal, "new recycling mechanism", is a suggesting idea to interpret the evolutive mechanisms of genoma, but the paper does not present a evidence of this fact.
Author Response
First, we would like to thank the reviewer for the enthusiasm demonstrated regarding our manuscript, and all the attention devote to it. In addition, his/her were important to now present an improved version.
In our opinion, the variety of works showing a not so strict organization of repetitive sequences, as tandem or dispersed repeats, highlight the need for a better classification, which may illustrate better the evolutionary relationship between them and allow their combined studied. However, in the process of revising the manuscript, we no longer think that a single-group classification is a true progress in the classification of the repetitive sequences. Thus, in this revised version of the manuscript, we propose a classification that is not based on the genomic organization of these sequences, but on the motifs conserved within them, more concretely in the presence or absence of specific retrotransposon/transposon domains. Thus, we propose two new groups for the repetitive sequence’s classification: repeats with retrotransposons/transposons specific domains (RTSD) or repeats with no retrotransposons/transposons specific domains (RTNSD).
For us, this new classification highlights more clearly the evolutionary relationship between these sequences, potentially inspiring more work addressing together sequences previously considered very distinct and unrelated. Their joint study is indeed very important for a better understanding of their function in distinct genomes, showing that the evolutionary relationship between them and the way they can convert into each other is highly associated with the evolution of the genomes themselves. According to that, we believe that future combined studies, regarding TEs and tandem repeats, namely concerning its chromosomal location and molecular similarity, will increase our knowledge on the evolution of eukaryotic genomes. In addition, the combined studies of related repetitive sequences can help us understand the reason for some evolutionary tracks and understand them, in a way that this genomic plasticity may have increased the adaptability of eukaryotic species.
Finally, all the works listed in table 1, of this revised version of the manuscript, support the DNA recycling mechanism (now named as DNA remodelling mechanism). The molecular dynamics of these sequences, which let them to evolve to a new sequence, occurs by mechanisms as unequal crossing-over that can result in chromosomal rearrangements, and certainly in genome’s evolution. Along the text we present evidences of that, namely in the point 3.1. Besides, the origin of new genes (or gene variants) from repetitive sequences could allow the genomes evolved in a way to be more adapted to the environmental conditions.
Round 2
Reviewer 3 Report
I propose the publication of the manuscript because the authors suggest a new interesting and well based hypothesis to classify the repeated sequences.